# Adaptive Coexistence of Synchronization and Anti-Synchronization for a Class of Switched Chaotic Systems

**Ling Ren *** and **Chenchen Peng**

School of Information and Control Engineering, Qingdao University of Technology, Qingdao 266520, China
* Correspondence: renling@qut.edu.cn

**Abstract:** This paper addresses the problem of coexistence of synchronization and anti-synchronization (CSAS) for a class of switched chaotic systems by adaptive control method, where the switched system is realized by unified chaotic systems under arbitrary switching signal. Firstly, necessary and sufficient conditions for the CSAS of the chaotic systems are proposed from two perspectives, one is by analyzing the parity of the system expression, and the other is by decomposing the system. Secondly, according to the obtained necessary and sufficient conditions, two algorithms are given to search the synchronization variables and anti-synchronization variables in the chaotic systems. Thirdly, the CSAS of the switched chaotic system can be achieved by a designed adaptive global controller with only one input channel under the arbitrary switching signal. Finally, the numerical simulation results verify the validity and effectiveness of the method we obtained.

**Keywords:** switched chaotic system; unified system; coexistence of synchronization and anti-synchronization; adaptive control method; common Lyapunov function

## 1. Introduction

Switched systems are a class of hybrid systems that arise in various fields of the real-life world. In the past two decades, increasing attention has been paid to the analysis and synthesis of switched systems due to their significance in both theory and applications, and many significant results have been obtained for the analysis and design of switched systems, such as electrical system [1–3], communication networks [4,5], flight control [6], network control systems [7–10], traffic control [11], and processes control [12,13], etc. Relatively speaking, chaos control has been investigated earlier. In the year of 1990 [14], the discovery of chaos synchronization by Pecora and Carroll greatly enhanced the research enthusiasm for the chaos control problem, such as the complete synchronization [15–19], anti-synchronization [20–23], phase synchronization [24–26], lag synchronization [27,28], generalized synchronization [29,30], projective synchronization [31–33], topological synchronization [34], etc. Simultaneously, lots of useful results have been obtained, and many chaos synchronization methods have been proposed by contemporary researchers, such as the linear feedback control [35,36], nonlinear control [37–39], active control [40,41], sliding mode control [42,43], etc. Coexistence of synchronization and ani-synchronization of the chaotic systems is a phenomenon that some variables in the master system synchronize the homologous variables in the slave system, and the others in the master system anti-synchronize the homologous variables in the slave system in the designed controller, which is an interesting phenomenon, while there are few works about this phenomenon. In our previous work [44], a specific necessary and sufficient conditions for the realization of CSAS and how to choose variables that can achieve anti-synchronization are investigated, and the adaptive control method is applied.

The chaotic system and switched system have been investigated comprehensively; however, the combination of the two, which is called switched chaotic system, is arousing the interest of researchers [45–50]. In [45], synchronization of the switched system is

investigated based on the Lyapunov method, and a sufficient condition is derived to ensure the synchronization between two switched systems. In [46], a sufficient condition for synchronization of identical master–slave switched nonlinear chaotic systems is obtained and is expressed in the form of bilinear matrix inequalities (BMIs). The switching control problem of a fractional-order chaotic system is studied in [47,48]. In [47], a novel fractional-order chaotic system which undergoes switching between integer-order and fractional-order dynamical subsystems with the help of the synchronization scheme is proposed, and this scheme is to enhance the level of security in chaotic communication. [48] proposes an interesting phenomenon that four fractional-order systems display a variety of nonlinear dynamical behavior. In [49], the problem of projective synchronization of chaotic systems and switched chaotic systems by adaptive control methods are addressed. In [50], chaotic heteroclinic networks with nontrivial intersections of stable and unstable manifolds as models of switching behavior in biological systems are proposed. The switched chaotic system is combined by the chaotic subsystems, compared with the single chaotic system, which is more complex. At the same time, the switched chaotic system has higher practical value, for example, in secure communication fields. The phenomenon of CSAS in the switched chaotic systems is a very interesting problem. While, to the best of our knowledge, such results for the CSAS of switched chaotic system does not yet exist.

Motivated by the above reasons, we investigate the CSAS for a class of switched chaotic systems. Necessary and sufficient conditions for the CSAS of the chaotic systems are proposed from two perspectives, one is by analyzing the parity of the system expression, the other is by decomposing the system. According to the obtained necessary and sufficient conditions, two algorithms are given to search the synchronization variables and anti-synchronization variables in the chaotic systems. Furthermore, the CSAS of the switched chaotic system can be achieved by a designed adaptive global controller with only one input channel under the arbitrary switching signal.

The contributions of this paper are mainly:

- Two necessary and sufficient conditions for the CSAS of the chaotic systems under a simple adaptive linear feedback controller are proposed.
- Based on the results above, two new algorithms are derived from searching the synchronization variables and anti-synchronization variables in chaotic systems.
- According to the results of the CSAS for chaotic systems, the problem of CSAS for the switched chaotic systems composed by the unified chaotic systems is investigated, and an adaptive global linear feedback controller with only one input channel is designed which can realize CSAS of the switched chaotic systems under the arbitrary switching law.

The rest of the paper is organized as follows. Section 2 presents the problem formulation. Some conceptions include synchronization, anti-synchronization, CSAS, and switched chaotic systems. Section 3 is divided into two subsections, for Section 3.1, the CSAS of a unified chaotic system is studied; for Section 3.2, we investigate the CSAS of switched chaotic system with arbitrary switching signal. In the end of every section, the numerical simulation is given to illustrate the effectiveness and correctness of the results we obtained. Section 4 is the conclusion of this paper and future works.

## 2. Preliminaries

Consider the following chaotic system described as

$$\dot{x} = f(x) \tag{1}$$

where $x \in \mathbb{R}^n$ is the state vector, $f(x)$ is smooth nonlinear function.

**Assumption 1.** $x_e = 0$ *is the equilibrium of the System (1), i.e.,* $f(x_e) = 0$.

**Remark 1.** *This assumption is a basic assumption in the system control theory. If $x_e \neq 0$ is the equilibrium of the System (1), we can make a transformation $x_t = x - x_e$, then for the new system $\dot{x}_t = f(x_t + x_e)$, $x_{te} = 0$ is also the equilibrium of the new system.*

Consider System (1) to be the master system, then the slave system can be drawn as:

$$\dot{y} = f(y) + u \tag{2}$$

where $y \in R^n$ is the state vector, $u$ is the controller to be designed.

Let $e = y - x$ be the synchronization error of the master and slave chaotic systems, and the synchronization error system is described as

$$\dot{e} = f(y) - f(x) + u = F(x,e) + u \tag{3}$$

Similarly, let $E = y + x$ be the anti-synchronization error of the master and slave chaotic systems, and the anti-synchronization error system is described as

$$\dot{E} = f(y) + f(x) + u = G(x,e) + u \tag{4}$$

**Definition 1** ([31]). *Considering the chaotic master System (1) and the slave System (2), define the synchronization error $e = y - x$ and the anti-synchronization error $E = y + x$. If and only if there exist a controller u simultaneously satisfies*

$$\lim_{t \to \infty} \|e_i(t)\| = 0,$$

*and*

$$\lim_{t \to \infty} \|E_j(t)\| = 0.$$

*where $e_i$ denotes the synchronization error of the $i-th$ variables in Systems (1) and (2), similarly, $E_j$ denotes the anti-synchronization error of the $j-th$ variables in system (1) and system (2), $i, j \in 1, 2, \cdots, n$ and $i \neq j$.*

*Then we say the master System (1) and the salve System (2) realize the coexistence of synchronization and anti-synchronization. That is to say, some variables in Systems (1) and (2) realize synchronization, and the other variables realize the anti-synchronization.*

*A switched chaotic system can be drawn as*

$$\dot{x} = f_{\sigma(t)}(x) \tag{5}$$

*where $\sigma(t) : [t_0, \infty) \to \Lambda = 1, 2, \cdots, m, \cdots, N$ is a piecewise right-continuous function, called the switching law. $\sigma(t) = m$ means the $m-th$ system is active. If $\dot{x} = f_m(x)$, $\forall m \in \Lambda$ denotes a chaotic system, then the System (5) is called switched chaotic system.*

**Remark 2.** *Throughout this paper, it is assumed that there are no jumps in the states at the switching instants, and that a finite number of switches occur on every bounded time interval. Let $x(t)$ denote the trajectory of the System (5) starting from $x_0 = x(t_0)$. In this paper, without any special instruction, the switching law $\sigma(t)$ is only related to the time t. In adddition, the master and slave systems simultaneously switch from one chaotic system to the other. The switching law is not predefined and is assumed to be arbitrary.*

Let System (5) be the master switched chaotic system, the corresponding slave switched chaotic system is given by

$$\dot{y} = f_{\sigma(t)}(y) + u_c. \tag{6}$$

where, the parameters are as the same as System (5) and $u_c$ is the controller to be designed.

Similar to Definition 1, the definition of CSAS for the switched chaotic systems is given.

**Definition 2.** *Considering the switched chaotic master System (5) and the switched chaotic slave System (6), define the synchronization error $e_{\sigma(t)} = y_{\sigma(t)} - x_{\sigma(t)}$, and the anti-synchronization error $E_{\sigma(t)} = y_{\sigma(t)} + x_{\sigma(t)}$. If and only if there exist a controller $u_c$ simultaneously satisfies*

$$\lim_{t \to \infty} \|e_{i\sigma}(t)\| = 0,$$

*and*

$$\lim_{t \to \infty} \|E_{j\sigma}(t)\| = 0.$$

*where $e_{i\sigma}(t)$ denotes the synchronization error of the $i-th$ variables in switched Systems (5) and (6), similarly, $E_{j\sigma}(t)$ denotes the anti-synchronization error of the j-th variables in switched Systems (5) and (6), $i, j \in 1, 2, \cdots, n$ and $i \neq j$.*

*Then we say the master System (5) and the slave System (6) realize the coexistence of synchronization and anti-synchronization.*

**Remark 3.** *The difference between Definition 1 and Definition 2 is that the investigated object of Definition 1 is the chaotic systems, while the Definition 2 studies the CSAS of the switched chaotic systems under the switching law $\sigma(t)$. Obviously, the problem of CSAS for the switched chaotic systems is much more complicated than that of the chaotic systems.*

**Lemma 1** ([15]). *For the chaotic System (1), if there exists a nonsingular coordinate transformation $z = Tx$ which can transfer the System (1) into*

$$\begin{cases} \dot{p}_1 = J_1(p_1, p_2), \\ \dot{p}_2 = J_2(p_1, p_2). \end{cases} \tag{7}$$

*where $p_1 = (z_1, z_2, \cdots, z_k)^T$, $k \geq 1$, $p_2 = (z_{k+1}, z_{k+2}, \cdots, z_n)^T$, and the following sub-system*

$$\dot{p}_2 = J_2(0, p_2). \tag{8}$$

*is asymptotically stable, then the stabilizing controller is designed as follows*

$$u = (u_1, u_2)^T = (k_1 p_1, 0)^T. \tag{9}$$

*where the feedback gain $k_1$ is updated by the following law*

$$\dot{k}_1 = -\gamma p_1^T p_1 = -\gamma \|p_1\|^2. \tag{10}$$

*and $\gamma$ is an arbitrary positive number. In other words, the following controlled system*

$$\begin{cases} \dot{p}_1 &= J_1(p_1, p_2) + k_1 p_1, \\ \dot{p}_2 &= J_2(p_1, p_2). \end{cases} \tag{11}$$

*is asymptotically stable, i.e., the System (11) is stabilized by the adaptive controller (9) with the adaptive gain (10).*

**Lemma 2** ([51]). *If the family of systems (5) has a common Lyapunov function (i.e., a positive definite radially unbounded smooth function V such that $\nabla V(X) f_{\sigma(t)}(x) < 0$ for all $x \neq 0$ and all $\sigma(t) \in \Lambda$), then the switched System (5) is asymptotically stable for any switching law $\sigma(t)$.*

## 3. Main Results

In this section, two subsections will be introduced to investigate the problem of the coexistence of synchronization and anti-synchronization (CSAS). Section 3.1 is the CSAS for the chaotic systems. Section 3.2 is the CSAS for the switched chaotic system under the arbitrary switching signal.

*3.1. Coexistence of Synchronization and Anti-Synchronization for Chaotic Systems*

In this subsection, necessary and sufficient conditions for the CSAS of chaotic systems are proposed from two perspectives. Theorem 1 is obtained by analyzing the parity of the system expression, and Theorem 2 is obtained by decomposing the system. Then according to the given two theorems, two algorithms are proposed to search the synchronization and anti-synchronization variables in the chaotic systems.

**Theorem 1.** *The master chaotic System (1) and slave chaotic System (2) realize CSAS if and only if $f(-x) = -f(x)$. i.e., $f(x)$ is the odd function.*

**Proof of Theorem 1.** The following is proof of sufficiency.

Since $f(-x) = -f(x)$, obviously, $e = 0$ and $E = 0$ are the equilibria of an unforced synchronization error system (3) and anti-synchronization error System (4), that is, $u = 0$, respectively. Then, if the controller $u = K(e)$ can stabilize synchronization error System (3), anti-synchronization error System (4) can be stabilized by the controller $u = K(E)$.

The following is proof of necessity.

If there exist the controllers $u = K(e)$ and $u = K(E)$ stabilizing error Systems (3) and (4), respectively, we can obtain that $e = 0$ and $E = 0$ are the equilibria of unforced synchronization error System (3) and anti-synchronization error System (4), respectively. Then, $f(-x) = -f(x)$.

That completes the proof. □

According to Theorem 1, we give the following Algorithm 1 to find the synchronous and anti-synchronous variables in the chaotic system.

---

**Algorithm 1:** *Distinguish the synchronous and anti-synchronous variables by analyzing the parity of the system expression.*

---

Step 1: Without loss of generality, we first select the variable $x_1$. If $f_1(x) = f_{11}(x_1)$ $+f_{12}(x_2; \cdots ; x_n)$ is an odd function, or $f_{11}(x_1) = \theta_1 x_1$, we can set $E_1 = x_1 + y_1$, where $\theta_1$ is a real number;

Step 2: If $f_2(x_2; \cdots ; x_n) = \theta_2 x_2 + f_{13}(x_3; \cdots ; x_n)$, we should set $E_2 = x_2 + y_2$, where $\theta_2$ is a real number. Else, if $f_2(x) = f_{21}(x_2) + f_{22}(x_1; x_3; \cdots ; x_n)$ is an odd function, or $f_{21}(x_2) = \theta_3 x_2$, we can set $E_2 = x_2 + y_2$, where $\theta_3$ is a real number. Then, we determine whether $E_2 = x_2 + y_2$ is suitable or not according to the condition that the origin is an equilibrium point of the anti-synchronization error system;

Step 3: When $i \leq n$, we can set $E_i = y_i + x_i$ or $e_i = y_i - x_i$ by the similar procedure in Step 2.

---

**Theorem 2.** *Considering the master chaotic System (1) and slave chaotic System (2), let $x = (w_m, z_m)^T \in R^n, w_m \in R^r, z_m \in R^{n-r}$. System (1) is described as*

$$\begin{pmatrix} \dot{w}_m \\ \dot{z}_m \end{pmatrix} = \begin{pmatrix} M(z_m)w_m \\ N(w_m, z_m) \end{pmatrix}. \tag{12}$$

*Accordingly, the slave chaotic System (2) can be described as*

$$\begin{pmatrix} \dot{w}_s \\ \dot{z}_s \end{pmatrix} = \begin{pmatrix} M(z_s)w_s \\ N(w_s, z_s) \end{pmatrix} + u. \tag{13}$$

*where $y = (w_s, z_s)^T \in R^n, w_s \in R^r, z_s \in R^{n-r}, u = (u_m, u_z)^T$ is the controller to be designed. If $N(-w, z) = N(w, z)$ is satisfied, then the master and slave system can be realized CSAS under the controller.*

**Proof of Theorem 2.** Let $E = w_m + w_s, e = z_s - z_m$, then

$$\dot{E} = \dot{w}_m + \dot{w}_s = M(z_m)w_m + M(z_s)w_s + u_w. \tag{14}$$

$$\dot{e} = \dot{z}_s - \dot{z}_m = N(w_s, z_m) - N(w_m, z_m) + u_z. \tag{15}$$

Due to $N(-w, z) = N(w, z)$, obviously, $e = 0, E = 0$ are the equilibrium of the uncontrolled master System (14) and the slave System (15) (i.e., $u = 0$), which satisfy Assumption 1. According to the nonlinear system control theory, there exists controller $u$, which can make sure the coexistence of synchronization and anti-synchronization for the master chaotic System (1) and slave chaotic System (2).

That completes the proof. $\square$

Consider master System (1) and slave System (2), and let

$$e = y - \beta x. \tag{16}$$

where $x, y$ and $e \in \mathbb{R}^n$ and $\beta = Diag(\beta_1, \beta_2, \beta_3, \cdots, \beta_n), |\beta_i| = 1, i \in \Lambda$ is a positive integer. Obviously, if $\beta_i = 1$ and $e_i = 0$ means the $i - th$ variable of master System (1) and slave System (2) realize synchronization. Similarly, if $\beta_i = -1$ and $e_i = 0$ means the $i$-th variable of master System (1) and slave System (2) realize anti-synchronization. So the error system is described as

$$\dot{e} = f(e + \beta x) - f(x) + u. \tag{17}$$

For the unforced error System (17), that is, $u = 0$. Solve the equation

$$f(\beta x) - \beta f(x) = 0, \tag{18}$$

we can get the solution $\beta_i$.

Firstly, we define two sets, for the first one:

$$\beta^{(i)} = \{\beta \mid There\ are\ exactly\ i\ elements\ of\ matrix\ \beta\ equal\ to\ 1\}.$$

where $1 \leq i \leq n - 2$. For example,

$$\beta = diag[1, -1, \cdots, -1] \in \beta^{(1)};$$

$$\beta = diag[-1, 1, 1, -1, \cdots, -1] \in \beta^{(2)}.$$

for the second one:

$$Q^{(j)} = \{There\ are\ exactly\ j\ equations\ of\ (18)\}.$$

where $2 \leq j \leq n - 1, j = n - i$. For example,

$$f_2(\beta x) - \beta_2 f_2(x) = 0\ and\ f_5(\beta x) - \beta_5 f_5(x) = 0.$$

which means there are two equations satisfy (18).

According to Theorem 2, we give the following Algorithm .

**Algorithm 2:** *Distinguish the synchronous and anti-synchronous variables by decomposing the system.*

---

Step 1: $i = 1$.

Check all elements of $\beta^{(i)}$ are the solutions of $Q^{(j)}$ or not, $2 \leq j \leq n - 1$. If one is yes, such as $\beta = diag[-1, 1, \cdots, -1]$ is a solution of $Q^{(j)}$, i.e., $Q^{(j)} = 0$, then set $w_m = [x_1, x_3, \cdots, x_n]^T$, $z = x_2$, thus the master system (1) is divided into the following two subsystems

$$\dot{x} = f(x) = \begin{pmatrix} \dot{w}_m \\ \dot{z} \end{pmatrix} = \begin{pmatrix} M(z)w_m \\ N(w_m, z) \end{pmatrix}.$$

where $w_m \in \mathbb{R}^{n-1}$, $z \in \mathbb{R}^1$, $N(w_m, z)$ is a nonlinear smooth function;

Step 2: $i = 2$.

Check all elements of $\beta^{(i)}$ are the solutions of $Q^{(j)}$ or not, $2 \leq j \leq n - 1$. If one is yes, such as $\beta = diag[-1, 1, 1, -1, \cdots, -1]$ is a solution of $Q^{(j)}$, i.e., $Q^{(j)} = 0$, then set $w_m = [x_1, x_4, \cdots, x_n]^T$, $z = [x_2, x_3]^T$, thus the master system (1) is divided into the following two subsystems

$$\dot{x} = f(x) = \begin{pmatrix} \dot{w}_m \\ \dot{z} \end{pmatrix} = \begin{pmatrix} M(z)w_m \\ N(w_m, z) \end{pmatrix}.$$

where $w_m \in \mathbb{R}^{n-2}$, $z \in \mathbb{R}^2$, $N(w_m, z)$ is a nonlinear smooth function;

Step 3: This procedure goes on until $i = n - 2$.

---

Next, the CSAS of unified chaotic system is investigated according to Theorem 1, Algorithm 1 and Theorem 2, Algorithm 2, respectively.

The unified chaotic system [46] is given by

$$\begin{cases} \dot{x}_1 = f_1(x) = (25\alpha + 10)(x_2 - x_1), \\ \dot{x}_2 = f_2(x) = (28 - 35\alpha)x_1 + (29\alpha - 1)x_2 - x_1 x_3, \\ \dot{x}_3 = f_3(x) = -\dfrac{1}{3}(8 + \alpha)x_3 + x_1 x_2. \end{cases} \tag{19}$$

where, the System (19) is chaotic for all values of the parameter $\alpha \in [0, 1]$. When $\alpha = 0$, the System (19) is the famous Lorenz system; when $\alpha \in (0, 0.8)$, the system is the general family of Lorenz systems; when $\alpha = 0.8$, it is called Lü system; when $\alpha \in (0.8, 1)$, the system is called general family of Chen system and when $\alpha = 1$, it is called Chen system.

Let System (19) be the master system, then the slave system is described as

$$\begin{cases} \dot{y}_1 = f_1(y) = (25\alpha + 10)(y_2 - y_1) + u_1, \\ \dot{y}_2 = f_2(y) = (28 - 35\alpha)y_1 + (29\alpha - 1)y_2 - y_1 y_3 + u_2, \\ \dot{y}_3 = f_3(y) = -\dfrac{1}{3}(8 + \alpha)y_3 + y_1 y_2 + u_3. \end{cases} \tag{20}$$

Now, the anti-synchronization variables will be found by Theorem 1 and Algorithm 1. According to Algorithm 1, considering the unforced slave System (20), we can set $E_1 = y_1 + x_1$ since $f_1(x) = (25\alpha + 10)(x_2 - x_1)$, and then we should set $E_2 = y_2 + x_2$. If we set $E_3 = y_3 + x_3$, then

$$\dot{E}_3 = -\frac{1}{3}(8 + \alpha)E_3 + x_1 x_2 + y_1 y_2 \tag{21}$$

$$= -\frac{1}{3}(8 + \alpha)E_3 + x_1 x_2 + (E_1 - x_1)(E_2 - x_2)$$

$$= -\frac{1}{3}(8 + \alpha)E_3 + E_1 E_2 - x_1 E_2 - x_2 E_1 + 2x_1 x_2.$$

It is clear that $E = 0$ is not an equilibrium point of the error System (21). In fact, the left hand side of the error System (21) equals zero. However, the right hand side of the error System (21) is not equal to zero, but equals to $2x_1x_2$. Therefore, we should set $e_3 = y_3 + x_3$, and obtain the error system given as (22).

Let $E_i = y_i + x_i (i = 1, 2)$, and $e_3 = y_3 - x_3$, so that the error system is obtained as

$$\begin{cases} \dot{E}_1 = f_1(x, y, e_3, E) = (25\alpha + 10)(E_2 - E_1), \\ \dot{E}_2 = f_2(x, y, e_3, E) = (28 - 35\alpha)E_1 + (29\alpha - 1)E_2 - E_1e_3 - x_3E_1 + x_1e_3, \\ \dot{e}_3 = f_3(x, y, e_3, E) = E_1E_2 - x_2E_1 - x_1E_2 - \frac{1}{3}(8 + \alpha)e_3. \end{cases} \tag{22}$$

where $E = (E_1, E_2)^T$.

Then, the anti-synchronization variables will be found by Theorem 2 and Algorithm 2. According to Equation (18), we obtain

$$f_1(\beta x) - \beta_1 f_1(x) = (25\alpha + 10)(\beta_2 - \beta_1)x_2 = 0,$$

$$f_2(\beta x) - \beta_2 f_2(x) = (28 - 35\alpha)(\beta_2 - \beta_1)x_1 + (\beta_2 - \beta_1\beta_3)x_1x_3 = 0,$$

$$f_3(\beta x) - \beta_3 f_3(x) = (\beta_1\beta_2 - \beta_3)x_1x_2 = 0.$$

i.e.,

$$\beta_2 = \beta_1, \tag{23}$$

$$\beta_2 = \beta_1 \text{ and } \beta_2 = \beta_1\beta_3, \tag{24}$$

$$\beta_1\beta_2 = \beta_3, \tag{25}$$

Noticing $n = 3$, according to Algorithm 2, we need only check $\beta^{(1)}$ are the solutions of $Q^{(2)}$ or not. In fact, there are exactly three elements of $\beta^{(1)}$. $\beta = diag(1, -1, -1)$, $\beta = diag(-1, 1, -1)$, $\beta = diag(-1, -1, 1)$. Obviously, $\beta = diag(-1, -1, 1)$ is a solution of $Q^{(2)}$, i.e., $\beta = diag(-1, -1, 1)$ is a solution of the Equations (23) and (24). In addition, there are no other elements of $\beta^{(1)}$ which are the solutions of $Q^{(2)}$. So there is only one solution of the CSAS for the unified system.

So set $w_m = (x_1, x_2)^T$, $z = x_3$, $w_s = (y_1, y_2)^T$, then according to Theorem 2, the unified chaotic master System (19) and slave System (20) can be transformed into

$$\begin{aligned} \dot{w}_m =& M(z)w_m = \begin{pmatrix} -(25\alpha + 10) & 25\alpha + 10 \\ 28 - 35\alpha - z & 29\alpha - 1 \end{pmatrix} w_m, \\ \dot{z} =& N(w_m, z) = -\frac{1}{3}(8 + \alpha)z + h(w_m), \\ \dot{w}_s =& M(z)w_s = \begin{pmatrix} -(25\alpha + 10) & 25\alpha + 10 \\ 28 - 35\alpha - z & 29\alpha - 1 \end{pmatrix} w_s + u. \end{aligned} \tag{26}$$

where $h(w_m) = x_1x_2$, $z$ is the coupling state vector.

The state vector $w_m$ and $w_s$ realize anti-synchronization, $x_3$ and $y_3$ realize synchronization under the controller $u = K(\cdot)$ and $K(0) = 0$.

Through the above analysis, we can see that the synchronization and anti-synchronization variables of the unified chaotic system found by Algorithm 1 and Algorithm 2 are consistent.

Next, the adaptive linear feedback controller will be designed.

Obviously, for the error System (22), if $E_2 = 0$, the sub-error system of (22) is

$$\begin{cases} \dot{E}_1 = -(25\alpha + 10)E_1, \\ \dot{e}_3 = -x_2E_1 - x_1E_2 - \frac{1}{3}(8 + \alpha)e_3. \end{cases} \tag{27}$$

is asymptotically stable regardless of the value of $\alpha$. So according to Lemma 1, the controller is designed as $u = (0, k_1 E_2, 0)^T$, where the feedback gain $k_1$ is updated as

$$\dot{k}_1 = -\gamma E_2^2 = -\gamma (y_2 + x_2)^2. \tag{28}$$

and $\gamma > 0$ is an arbitrary number.

**Theorem 3.** *Considering the master unified chaotic System (19), if the controller $u = (0, k_1 E_2, 0)^T$ is added into the slave chaotic System (20), where the feedback gain $k_1$ is updated as (28), then the error System (22) is asymptotically stable, i.e., the master unified chaotic System (19), and the slave chaotic System (20) can realize CSAS.*

**Proof of Theorem 3.** For the error System (22) with the controller $u = (0, k_1 E_2, 0)^T$, we introduce the following positive definite Lyapunov function

$$V(E, e_3, k_1) = \frac{1}{2}(e_3^2 + E^T E) + \frac{1}{2\gamma}(k_1 + L_1)^2. \tag{29}$$

where

$$L_1 > M \sup_{E_2 \neq 0} \frac{e_3^2 + E^T E}{E_2^2},$$

and

$$M = \max_{i=1}^{3} f_i(x, y, e_3, E).$$

Differentiating the Lyapunov function V along the trajectories of the augment system, we obtain

$$
\begin{aligned}
\dot{V} &= E_1 \dot{E}_1 + E_2 \dot{E}_2 + e_3 \dot{e}_3 + \frac{1}{\gamma}(k_1 + L_1)\dot{k}_1 \\
&= E_1 f_1(x, y, e_3, E) + E_2(f_2(x, y, e_3, E) + k_1 E_2) + e_3 f_3(x, y, e_3, E) - (k_1 + L_1)E_2^2 \\
&= E_1 f_1(x, y, e_3, E) + E_2 f_2(x, y, e_3, E) + e_3 f_3(x, y, e_3, E) - L_1 E_2^2 \\
&= \sum_{i=1}^{3} E_i f_i(x, y, e_3, E) - L_1 E_2^2 \\
&\leq M(E_1^2 + E_2^2 + e_3^2) - L_1 E_2^2 < 0.
\end{aligned}
$$

According to the Lyapunov stability theory, the error Systems (22) is asymptotically stable. That is to say, the master System (19) and the slave System (20) realize CSAS with the designed controller, which has only one input channel. That completes the proof. $\square$

**Remark 4.** *From Theorem 3, we can see that, no matter what is the switching law $\sigma(t)$, the designed controller keeps the same, i.e., the controller structure does not need to be changed whenever the chaotic systems switch. Moreover, the designed controller has only one input channel.*

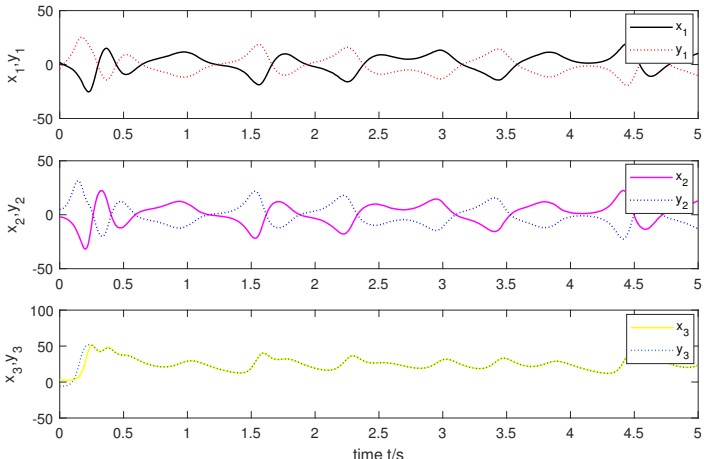

**Figure 1.** States of CSAS for the unified chaotic systems ($\alpha = rand$).

Next, we give numerical simulations to illustrate the efficiency of the designed controller. The initial values are given as $x(0) = (2, -2, 3)^T$, $y(0) = (-4, 5, -6)^T$, and $k_1(0) = -1$. The simulation results are shown from Figures 1–5.

In Figures 1 and 2, $\alpha = rand$, which denotes $\alpha$ is random and $\alpha \in [0, 1]$. Figure 1 shows that the state variables $x_1$ and $x_2$ of the master system are anti-synchronized with $y_1$ and $y_2$ of the slave system, respectively. Figure 2 shows that the CSAS error system is asymptotically stable, and the adaptive feedback constant $k_1$ converges to a fixed negative constant.

Especially, the simulation results are also given to illustrate the efficiency of the designed controller when $\alpha$ is given a certain value. Figures 3–5 show the CSAS phenomenon when $\alpha = 0$ (Lorenz system), $\alpha = 0.8$ (Lü system) and $\alpha = 1.0$ (Chen system). The simulation results show that even if the valve of $\alpha$ is changed, the designed controller is also effective. It further shows that the controller we designed is a global controller.

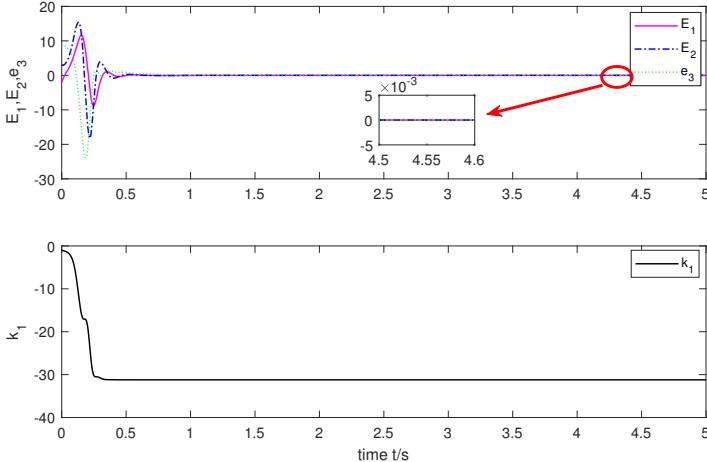

**Figure 2.** CSAS error for the unified chaotic systems and adaptive law of $k_1$.

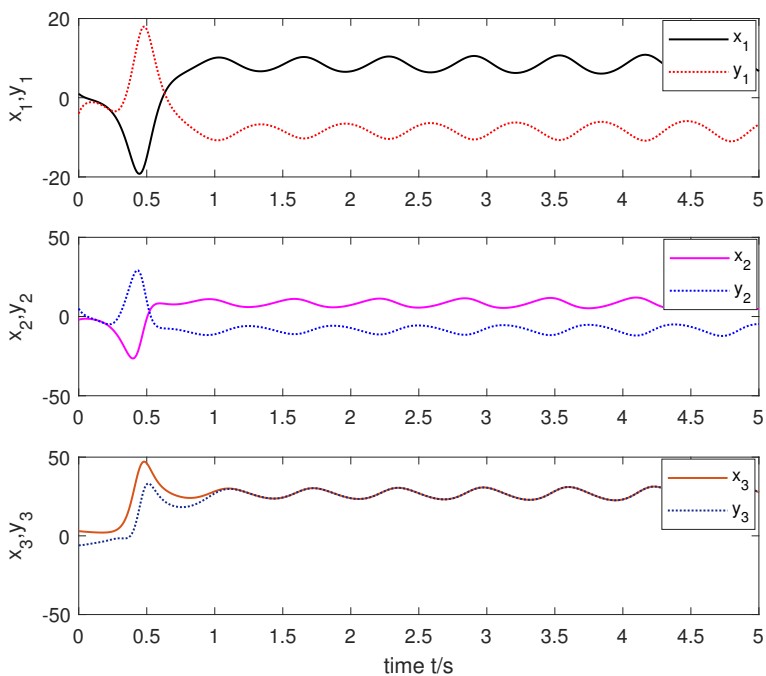

**Figure 3.** States of CSAS for the unified chaotic systems ($\alpha = 0$).

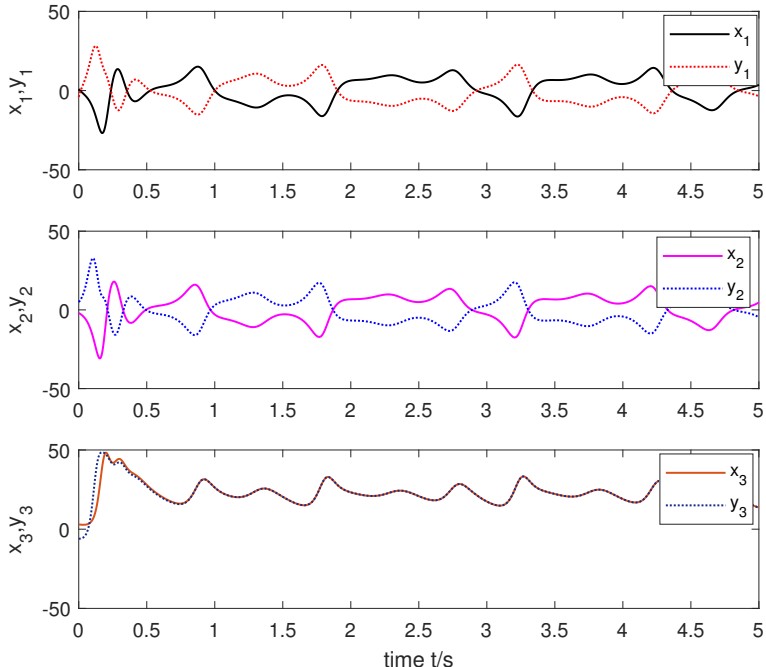

**Figure 4.** States of CSAS for the unified chaotic systems ($\alpha = 0.8$).

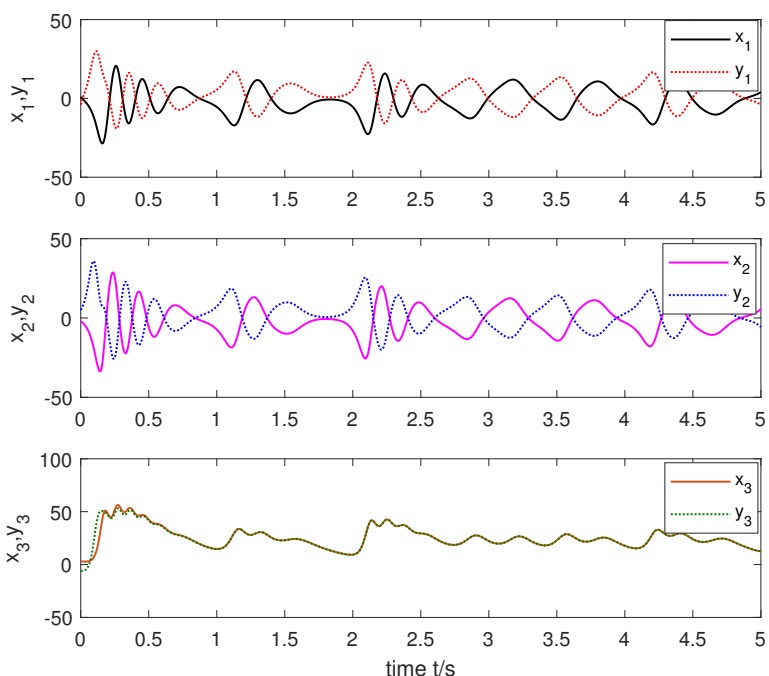

**Figure 5.** States of CSAS for the unified chaotic systems ($\alpha = 1.0$).

### 3.2. Coexistence of Synchronization and Anti-Synchronization for Switched Chaotic Systems

If sudden switching in the parameter $\alpha$ is considered in the unified chaotic System (19), the obtained system can be viewed as a switched chaotic system [46]. Figure 6 shows the switching instants of the parameter $\alpha$, so that the unified chaotic system switches among the Lorenz–Chen-Lü family. In fact, these three subsystems in the unified chaotic system can be switched in different combinations to produce different switched chaotic systems. In the contemporary literature, synchronization and control performance are studied among various classes of chaotic and hyperchaotic systems [45,52]. To our best knowledge, there is no work about designing a global controller to guarantee CSAS among arbitrary switching laws of the unified chaotic system family. Motivated by Pan et al. [46], Figure 6 shows that the piecewise continuous switching law $\sigma(t)$ changes the value of $\alpha$ in the unified chaotic system model of (19) thereby switching between different chaotic models. Thus, $\alpha \in [0, 1]$, which is the range where $\alpha$ can vary in the unified chaotic System (19). For example, the system behaves like either of the Lorenz–Lü-Chen system, or the system evolves as a different chaotic system: Lorenz–Chen-Lü system. In fact, there are six switching orders. Such a configuration makes the whole switched chaotic system a combination of three chaotic subsystems, which is more complex than the usual ones. In this paper, the master and slave systems are assumed to be driven by the same switching law.

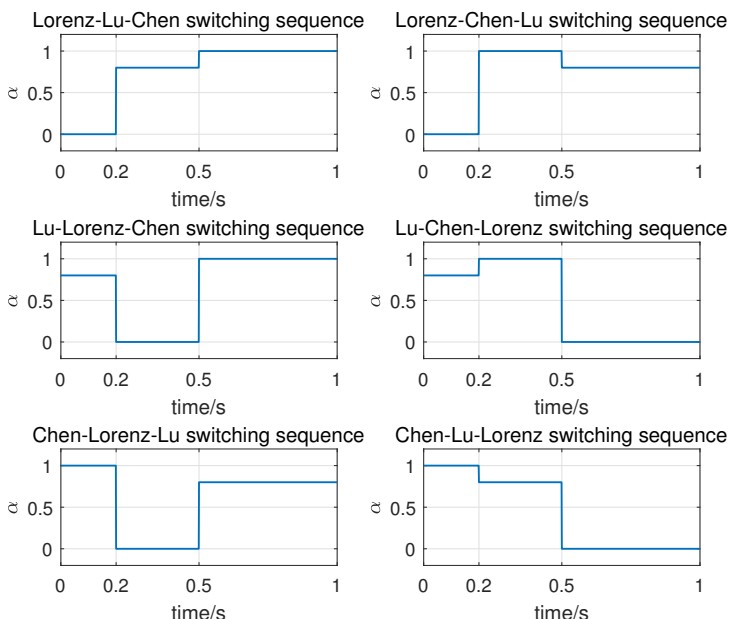

**Figure 6.** Switching law for the switched chaotic systems.

Consider the unified chaotic System (19), when $\alpha = 0$, it is called Lorenz system; when $\alpha = 0.8$, it is called Lü system; when $\alpha = 1$, it is called the Chen system. In this part, the switched chaotic System (5) is comprised of these three chaotic systems, i.e., $N = 3$. $i = 1$ denotes the Lorenz system; $i = 2$ denotes the Lü system; $i = 3$ denotes the Chen system. If $\sigma(t) = 1$, it means Lorenz system is active; if $\sigma(t) = 2$, it means Lü system is active; if $\sigma(t) = 3$, it means Chen system is active.

**Theorem 4.** *Consider the switched chaotic master System (5) and switched chaotic slave System (6), where the switching law is chosen as $\sigma(t) = 1, 2, 3$, the chaotic system '1' is Lorenz system, '2' is Lü system, '3' is Chen system, if a global adaptive controller is designed as*

$$u_c = (0, k_1 E_2, 0)^T = (0, k_1(x_2 + y_2), 0)^T, \tag{30}$$

*where the feedback gain $k_1$ is updated as*

$$\dot{k}_1 = -\gamma E_2^2 = -\gamma(y_2 + x_2)^2. \tag{31}$$

*and $\gamma > 0$ is an arbitrary number.*

*Then the state variables $(x_1, y_1)$ and $(x_2, y_2)$ of the switched chaotic system realize synchronization respectively; simultaneously, $x_3$ and $y_3$ realize anti-synchronization under arbitrary switching law $\sigma(t)$.*

**Proof of Theorem 4.** According to Theorem 3, Lyapunov function (29) is suited to the unified chaotic systems for $\alpha \in [0, 1]$, so Lorenz system ($\alpha = 0$), Lü system ($\alpha = 0.8$) and Chen system ($\alpha = 1$) are three chaotic systems among them. That is to say, the Lyapunov function (29) is a common Lyapunov function for the Lorenz system, Lü system and Chen system. According to Lemma 2, Lorenz system, Lü system, and Chen system can be asymptotically stable for any switching law $\sigma(t)$.

That completes the proof. $\square$

To illustrate the efficiency of the proposed adaptive controller for the switched chaotic systems, a numerical examples will be given.

Switched chaotic system is comprised by Lorenz system, Lü system and Chen system, and the switching law is chosen as

$$\sigma(t) = \begin{cases} 1, & t \in [t_{3m}, t_{3m+1}), \ t_{3m+1} - t_{3m} = rand \\ 3, & t \in [t_{3m+1}, t_{3m+2}), \ t_{3m+2} - t_{3m+1} = rand \\ 2, & t \in [t_{3m+2}, t_{3m+3}), \ t_{3m+3} - t_{3m+2} = rand \end{cases} \tag{32}$$

where, $m = 0, 1, 2, \cdots$, is a non-negative integer, '1' denotes the Lorenz system; '2' denotes the Lü system; '3' denotes the Chen system. '*rand*' denotes the active time of every subsystem being arbitrary.

According to the switching law (32), the initial values are given as $\alpha = 0$, denotes the first active subsystem is Lorenz chaotic system, $x(0) = (2, -2, 3)^T$, $y(0) = (-4, 5, -6)^T$, and $k_1(0) = -1$. The simulation results are shown in Figures 7 and 8.

Figure 7 shows that the state variables $x_1$ and $x_2$ of the master system are anti-synchronized with $y_1$ and $y_2$ of the slave system, respectively. Figure 8 shows that the CSAS error system is asymptotically stable under the designed controller with the arbitrary switching law $\sigma(t)$ and the adaptive feedback constant $k_1$ converges to a fixed negative constant.

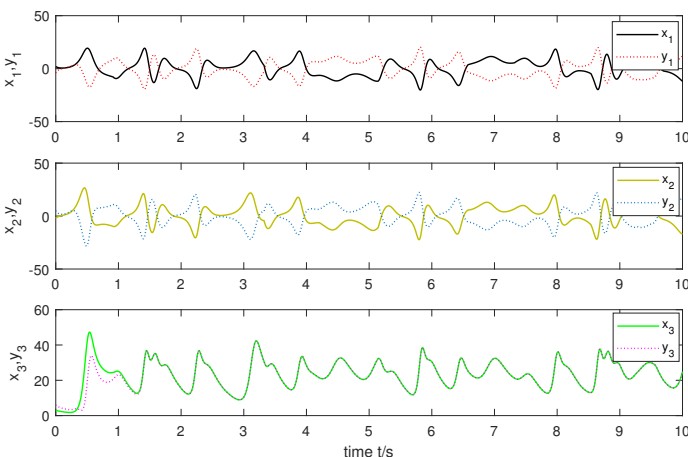

**Figure 7.** States of CSAS for the switched chaotic systems.

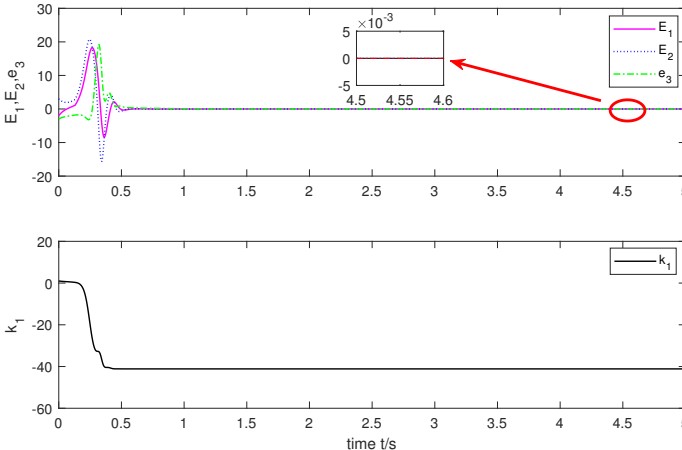

**Figure 8.** CSAS error for the switched chaotic systems and the adaptive law of $k_1$.

## 4. Conclusions and Future Works

In conclusion, this paper investigated the problem of coexistence of synchronization and anti-synchronization for chaotic systems and switched chaotic systems by adaptive

control method, which is a more complicated phenomenon than synchronization. Necessary and sufficient conditions are given to judge whether the chaotic system can realize CSAS. Moreover, two algorithms were proposed to look for the variables in the chaotic system which can achieve anti-synchronization, and the others realize synchronization. Then, we studied the CSAS of the switched chaotic system under the designed adaptive controller with the arbitrary and the switching signal, where the switched system was comprised of a unified chaotic system. It is important to note that the adaptive global controller has only one input channel, which is easily realized in fact.

As can be seen, the switching signals are related only with time, so in future work, some different types of switching signals can be investigated. Moreover, the CSAS of the switched chaotic system is realized in theory and simulation, and circuit implementation and secret communication may be valuable study directions.

**Author Contributions:** Conceptualization, L.R.; methodology, L.R. and C.P.; software, L.R.; validation, L.R. and C.P.; formal analysis, C.P.; investigation, L.R.; resources, C.P.; data curation, L.R.; writing—original draft preparation, L.R.; writing—review and editing, L.R.; visualization, C.P.; supervision, L.R.; project administration, L.R.; funding acquisition, C.P. All authors have read and agreed to the published version of the manuscript.

**Funding:** This research was supported by the National Natural Science Foundation of China (62203247).

**Data Availability Statement:** Not applicable.

**Conflicts of Interest:** The authors declare no conflict of interest.

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
