# Peer review of "Adaptive Coexistence of Synchronization and Anti-Synchronization for a Class of Switched Chaotic Systems"

_processes, doi:10.3390/pr11020530_

Round 1

Reviewer 1 Report

In this paper, the authors design a controller that achieves coexistence of synchronization and anti-synchronization (CSAS) for switched chaotic system. Then, they investigate its validity. I think that the paper contains some new materials worth for publishing in ``Processes’’ journal. However, there are a few points that are not clear. I made some comments on the manuscripts.

1. In page 7, does ``rand’’ indicate random? If so, what does ``alpha=rand’’ mean? If the initial alpha is determined by rand, how are the other alphas determined?

2. What is the value of alpha used in Figures 1 and 2? Is there any evidence that CSAS can be observed even if alpha changes?

3. In section 3.2, the authors similarly describe ``alpha=rand’’. From equation (32), it appears that the time at which α switches is random. The fact that the alpha value switches between 0,0.8,1.0 depending on time and the fact that ``alpha=rand’’ seems contradictory. Is only the initial alpha determined by rand and the others by equation (32)?

4. There are several formatting discrepancies in the paper. Authors should correct them to avoid reader confusion. For example, R formatting, beta subscripts, etc.

5. There are small typos. In page 3, line 2, sigma is not in subscript. In the proof of theorem 7, one of the brackets for eq. (4) is missing.

Reviewer 2 Report

Reviewer’s comments

In the paper entitled, ‘Adaptive coexistence of synchronization and
anti-synchronization for a class of switched chaotic systems
’, the problem of coexistence of synchronization and anti-synchronization (CSAS) for a class of switched chaotic systems is addressed by adaptive control method..

In my view, the novelty in the paper is not satisfactory, but can be enhanced by incorporating the following mandatory changes:

1)      The paper has a lot of typos and grammatical errors. The writing style needs to be improved. For instance,

                    i.             “…While, To...

                  ii.             “…have not been existed…”

                iii.            Typos like “…salve…” in Definitions 3 and 4.

                iv.            “…sufficiency proof …”

                  v.            “…and Chen system are as three chaoti …”

                vi.            “…system are anti-synchronize with…

              vii.            “…which a more complex…” and many more.

2)      The theorems 8 and 10 are missing. Algorithms 1-7 are missing..

3)      The authors’ claim “…The chaotic system and switched system have been investigated comprehensively, however, the combination of the two,
which is called switched chaotic system, relatively speaking, is rarely investigated
…” The authors may enhance their literature survey since switched chaotic systems have been extensively reported in literature as in follows.

                    i.            Fractional-order systems with diverse dynamical behaviour and their switching-parameter hybrid-synchronisation”, European Physical Journal Special Topics, doi: https://link.springer.com/article/10.1140/epjst/e2018-00063-9

                  ii.            Switching synchronisation control between integer-order and fractional-order dynamics of a chaotic system, https://ieeexplore.ieee.org/abstract/document/7846517

                iii.            Chaotic heteroclinic networks as models of switching behavior in biological systems, https://doi.org/10.1063/5.0122184

4)      The switching laws show that the switching is primarily done during the transient phase between 1-3 s as in (32) and Fig 3. It would be interesting to observe the switching of the parameter alpha after the system reached steady state.

5)      The authors may cite physical applications where specifically their switching of the chaotic systems is applicable and how.

All in all, I suggest a major revision for the paper, incorporating all the above comments. The paper should be reviewed again after the above revisions.

Reviewer 3 Report

This paper addresses the problem of coexistence of synchronization and anti-synchronization (CSAS) for a class of switched chaotic systems by adaptive control method. . For this paper, I have the following comments.

1.     The research motivation and contribution of this paper are not clearly given.  The author needs to clearly provide the research motivation and advantages of this paper compared with the existing methods.

2.     Some variables are not defined like e_i E_j in definition 3, Beta_i in page 4

3.     Definition 4 needs a reference.

4.     Compared with the existing results, what additional difficulties do you meet? How do you overcome those difficulties?

5.     Maybe some more remarks after the development of the main results would be helpful.

6.     The English should be further polished, and some sentences should be written in a better format.

In page 2 ‘In this section, 2….’ Should be  In this section, two….

The following sentence is not clear “In this subsection, two theorems are given. Theorem 7 prescribes the necessary and sufficient condition for the CSAS of the chaotic systems from the perspective of the mathematical expression of the system, and the Theorem 9 is through the decomposition of the system to give the necessary and sufficient condition for the CSAS of the chaotic systems. Then according to the given two theorems, two algorithms are given to search the synchronization variables and anti-synchronization variables in the chaotic systems.”

In my view, this paper should be profoundly improved and I recommend that it be rejected.

Round 2

Reviewer 2 Report

Can be accepted with minor changes in English writing

Reviewer 3 Report

Even though the paper is improved compared to the first version, there are many issues to consider. 

1/ Many English typos have to be fixed. Example page 6 line 57, the contributions of this paper is (are) mainly:

2./ All variables in the text must be defined. (like x_te in page 2 line 80, \Lambda in page 6 line 180).

In page 5, I think Theorem 1 is a fact not theorem.

3/Inpage 6, it iswriten ''firstly we define two sete''. where is the second set?

4/ In page 6, 'it is written k=0,1. What is it k?

5/ In page 6, \beta5 and f_5 are not defined.

6/ What is it Q(j) in page? What is the difference between \beta_i and \beta^(i)?

7/What means rand in the paper?

8/How to check the stability of the switched system for an arbitrary signal?

9/ A comparison with existing results [45-47] is compulsory.

Round 3

Reviewer 3 Report

I have checked the paper. The authors have responded to my comments